# Eat to Live, Don’t Live to Eat: Black Men, Masculinity, Faith and Food

**DOI:** 10.3390/ijerph17124264

**Published:** 2020-06-15

**Authors:** Letisha Engracia Cardoso Brown

**Affiliations:** Department of Sociology, Virginia Tech, Blacksburg, VA 24060, USA; letishab@vt.edu

**Keywords:** Black men, masculinity, faith, food, health

## Abstract

Men often have poorer health outcomes than women. In the United States, Black men in particular tend to have worse health than not only Black women but other racial/ethnic groups of men. One factor that contributes to health is the role of masculinity. Previous research notes that men who cling to hegemonic notions of masculine identity tend to engage in negative health behaviors. However, hegemonic masculinity is not the realm in which Black men exist. Criminalized, surveilled, and subject to structural racism and racial discrimination, Black masculinities exist on their own spectrum separate from that of White men. One characteristic associated with Black masculinity is that of faith, and faith is a growing field of study with respect to health. This paper examines the relationship between Black masculinity as framed by faith in shaping the food and eating habits of Black men. Food and eating are central to health and well-being yet remain understudied with respect to Black masculinity through the lens of faith. This study offers a qualitative account of Black men’s experiences through the use of in-depth interview data. The key finding of this study is that fasting operates as a mechanism of health promotion for Black men. This paper utilizes the term Black men as an all-encompassing term of members of the African diaspora as opposed to African American in order to recognize the diversity of the participants in this study.

## 1. Introduction

Masculinity refers to the social construction of what it means to be a “man” in a particular social context [1,2]. Taken this way, masculinity is fluid and adaptable with respect to history, culture, and time [1,2]. In western societies such as the US, traditional, that is to say hegemonic masculinity, is associated with traits such as competitiveness, emotional detachment, dominance, autonomy, being physically tough, and heterosexuality [1,2,3,4,5]. Constructing masculinity this way within western society, means that it is not the purview of all men. Rather, as scholars such as Genoe and Singleton (2006) note, hegemonic masculinity within the West belongs to those who are “Caucasian, heterosexual, middle class, and in their early midlife,” [2]. Though often the top of the racial hierarchy within the West, it has been argued that strong adherence to hegemonic forms of masculinity places men at a disadvantage through its restrictive nature [5]. Within the context of health in particular, hegemonic masculinity has been discussed as shaping various health behaviors including eating habits, recovery from injury, exercise, and the deterioration of one’s mental state [2]. Furthermore, research contends that men face qualitatively different health disparities compared to women including higher mortality rates from 12 of the 15 leading causes of death, lower life spans overall, and a greater lifetime risk of developing certain cancers [5,6].

In order to better understand these processes, it is necessary to consider the role of social and cultural factors play on shaping these behaviors [5]. Though earlier scholarship has focused on hegemonic forms of masculinity, contemporary research is paying attention to the role that social characteristics such as race, ethnicity, class, and sexuality play in how men construction their own masculine ideals [2,7]. This is especially important for Black men whose health continues to be worse than nearly all other racial/ethnic groups in the United States [8,9]. This paper utilizes the term Black, as opposed to African American as an inclusive term for members of the African diaspora. Though the majority of the participants in this study identify as African American, there are participants who were born outside of the US and/or have roots beyond the United States. From a historical standpoint, the life expectancy of Black men in the United States has consistently been lower than that of whites and other racial/ethnic groups of men [8,9]. The leading causes of death among Black men include diabetes, heart disease, stroke, unintentional injuries, kidney disease, cancer, chronic lower respiratory disease, and homicide [10]. As to why Black men continue to experience poorer health needs continued exploration and is in the process of expanding [8,9].

One issue that has been discussed when it comes to the study of Black men and health is that it is often contextualized through a lens of racial/ethnic health disparities [9]. Though racial and ethnic health disparities are an important avenue of exploration, what is often missing from examination is the role that gender plays in the lived experiences of Black men when it comes to shaping their health and well-being [8,9]. A second issue that arises is the comparative nature of research on Black men’s health that juxtaposes them against either Black women or other racial/ethnic groups of men [8,9]. Black men exist in a unique space with respect to social norms, physical environment, socioeconomic circumstances, and cultural patterns that shape their health and well-being [9]. Research that focuses exclusively on Black men then can highlight the specific mechanisms and pathways that influence their experiences of health. One area of study that is expanding when it comes to the study of health is that of the role of religion [11,12,13,14]. Religion remains an influential part of Black communities [15,16] and thus within the lives of Black men. Current research on religion and health has examined how religious involvement tends to contribute to healthier lifestyles [12,13,17]; however, focus on food and eating with respect to religion and health requires further exploration.

The goal of this paper is to examine how adherence to aspects of Black masculinity including faith (contextualized as spiritual religiosity) shape Black men’s experiences of food and eating. Food and eating shape overall health and well-being [18] and play a role in religion as well. Examining the role that faith has on food can provide new insight into underlying mechanisms tied to health and well-being. Furthermore, by using Black masculinity as a lens, this paper will contribute to the growing literature on Black men and health. Thus this paper provides a qualitative account of food and eating as driven by faith-based experiences (predominantly Christian-based beliefs) and notions of Black masculinity. First, this paper provides an overview of Black masculinity, followed by a discussion of the status of Black men and health. Next, this paper examines the relationship between health and faith, followed by a discussion of faith, food, and health. Following a discussion of the methodological considerations, the qualitative analyses offer narrative accounts of Black men’s experiences of food and eating with respect to faith. Understanding such processes can lend insight into the overall health and well-being of Black men in the United States.

### 1.1. Black Masculinity

Black men experience socialization differently than their White male counterparts; as such, Black men experience a different social reality and thus embody different forms of masculine identity [8,19,20]. Historically, Black masculinity was framed as a “problem,” (not just to the Black family but to the nation as a whole), and thus, Black men were framed as troubled or deviant [7] with research suggesting that Black men were sexually permissive (hypersexual) [21,22] and failed to provide for their families [23]. These early conceptualizations of Black masculinity were shaped largely by racist understandings of Black men tied to the legacies of slavery and Jim Crow in particular. Later discussions of Black masculinity still focused on its supposed dysfunctionality but also proposed that Black men enacted certain behaviors as a means of pushing back against racist oppression [7,19,24]. One means that Black men have used to push back against racist assumptions is the notion of “the cool pose” [19]. The “cool pose” functions as a “ritualized form of masculinity that entails behaviors, scripts, physical posturing, impression management, and carefully created performances to deliver a single, critical message: pride, strength, and control” (p. 4). In a society in which Black men are perpetuality criminalized and dehumanized, the “cool pose” offers an alternative form of Black masculinity. However, while the critical message of “pride, strength and control” pushes back against ideas of Black men as being uncontrollable “bucks” [22], “posturing” suggests a certain inauthenticity [7]

Research conducted on Black men’s own conceptualizations of their manhood paint a slightly different picture. Studies of urban Black men conducted by Hunter and Davis [25,26] noted four domains of Black masculine identity: (1) self-determination and accountability, (2) family, (3) pride, and (4) spirituality and humanism. Such conceptualizations push back against ideas of Black men as absent fathers [23] or as thugs [22]. The work by Hammond and Mattis 2005 built upon this earlier research, showcasing overlap as well as expansions. While spirituality-religiosity, accountability, and family centeredness were revealed by the Black men in their study, so too were characteristics such as leadership and guidance, community involvement, and more [7]. Black masculine identity is complex and must be contextualized by the ways in which Black men are regarded as racialized, gendered, and sexualized human beings [22,27]. Research on Black masculinity that does not recognize the matrix of domination [27] under which Black men live, misses the unique position in which Black men’s experiences are situated. Black men’s experiences should not be considered additive or comparative to those of Black women but as their own set of unique circumstances that shape their lived experiences; including that of food and eating and overall health and well-being.

### 1.2. Black Men and Health

Of all racial/ethnic groups in the US, Black men continue to have the shortest life expectancy [9,20]. Furthermore, when it comes to the leading causes of death and disability within US society, Black men suffer disproportionately [8]. Specifically, chronic health conditions such as heart disease, hypertension, prostate cancer, and diabetes are issues that impact Black men’s rate of mortality, which are often higher than those of other race/gender groups [8]. Research on Black men and health has highlighted these issues, as well as the fact that Black men have a disproportionately high incidence as well as prevalence of dying from HIV/AIDS, particularly among young Black men [8]. Men in general experience poorer health than women, yet the experiences of Black men highlight the importance of examining health disparities with an intersectional lens [8]. Though research has identified behavioral factors that partially explain gendered differences in health, research that emphasizes the experiences of Black men in particular merits further investigation. In general, men are said to engage in behaviors such as smoking, substance abuse, violence, and unsafe driving at higher rates than their female counterparts. However, research also contends that Black men are *less* likely to participate negative health practices such as smoking and excessive drinking [20]; however, the negative health consequences of such behaviors are higher for Black men in the US [20].

There is no denying that Black men in America are subjected to experiences and expectations that vary from those of other racial/ethnic groups (well as Black women), and that many of these experiences negatively impact their health. Research highlights that Black men are more likely to live in poverty, which may explain health disparities as a result of differences in income as well as wealth [28,29]. However, even for middle class Black men, health disparities remain [28]. Beyond the experiences of diminished returns with respect to income and wealth, Black men are also exposed to heightened policing and criminalization as a result of racist attitudes and practices that contribute to an increase in social and physiological stressors that negatively impact health [8]. Black men in America are socialized with respect to race, gender, class, and more within the matrix of social categories that drive life chances, thereby making their experiences qualitatively different from those of members of other racial/ethnic as well as gendered groups [8,9,20].

The study of faith and health continues to grow. Considering the importance of faith within Black communities, it is an ideal avenue for the exploration of Black men’s health in particular. Though faith is a dominant aspect of Black masculine identity [7,25,26], Black men continue to suffer disproportionately from negative health outcomes. Thus, an in-depth look at faith as an avenue of health promotion through qualitative inquiry warrants close attention.

### 1.3. Faith and Health

Faith has many meanings within society. Over the past several decades studies have highlighted the ways that religious involvement conceptualized as “observable feelings, beliefs, activities, and experiences in relation to the spiritual, divine, or supernatural” (p. 239, [12]) contributes to better health outcomes such a lowered risk of mortality [12,13]. Often, the terms, religion, spirituality, and faith are used interchangeably within health literature [30]. Within the context of this paper, I will rely on the term “faith” to discuss spiritual-religiosity as “seeking guidance about one’s behavior from God, a higher power, spirit, religious texts, or leaders; having belief in, relationship with or connection with any of the entities referenced above” [7]. This definition of faith stems from the work of Hammond and Mattis (2005) that examined how Black men discussed their conceptualization of manhood. As discussed previously, Black men’s experiences of manhood are qualitatively different than those of White men in the US [7]., Black men seek to “reconstruct traditional markers of masculinity” [7] (p. 124) by providing for others in a myriad of ways including through the practice of faith [7].

Research should thus examine the role that faith-based activities may contribute to health promotion of such groups but go beyond the examination of smoking and drinking. While both contribute to overall health, when it comes to the examination of religion and health, these behaviors have been studied. What has been less studied is the experiences of faith-based practices in relation to the experience of food and eating. Like smoking and drinking, food behaviors also matter for overall health [31]. One aspect of food and eating that should be examined when it comes to faith-based practices and health is the notion of fasting. Fasting is intimately connected to both health and faith yet remains an understudied phenomenon when it comes to the experiences of Black men and their attitudes and health-based behaviors. Though such a practice through the lens of faith cannot account for the experiences of structural racism and racial discrimination within health care contexts, understanding how Black men discuss their relationship with faith and food can shed light on how they perceive and experience their own health and well-being.

### 1.4. Faith, Food, and Health

How and what we eat is shaped by a myriad of societal factors. For instance, how and what we eat is shaped by our economic circumstances, race, age, and health status [18,32]. Furthermore, as Dallam (2014) notes, “Entire cultural groups have explicit rules about foods, and these may include ideas about what foods are encouraged and what foods are taboo, what materials count as food, what specific foods are consumed during rituals and holidays, what kinds of food combinations are acceptable and forbidden how foods must be prepared and what foods must be eaten” (xvii). When faith is added to the mix, what and how individuals eat becomes even more complicated [32]. According to Dallam (2014), faith and food intersect in many was as several “religions have sacred rituals involving both special foods and acts of consumption” (xviii). Research that examines the relationship between faith and food has the potential to shed light on overall health and well-being.

When it comes to Black communities, food remains a large part of faith culture [33]. Specifically, Hicks (2004) contends that “food and religious identity are symbolic in nature,” (p. 144), and when it comes to “many Black Baptist churches for example, a Sunday that does not culminate in a meal would be for some sacrilegious” (p. 144). Thus, for some faith-based cultures within the Black community, the Sunday meal is “seen as a primary time for fellowship, and thus a continuation of ministry and worship,” (p. 144). Within these spaces, faith and food function as an aspect of community support, which is beneficial to overall health and well-being. Examining the role of faith on Black men’s experiences of food and eating can illuminate underlying mechanisms that contribute to their overall health.

### 1.5. The Present Study

The purpose of the present study is to examine the role that aspects of Black masculinity (faith in particular) play in their lived experiences of food and eating. Black men consistently have worse health outcomes than their other racial/ethnic peers; yet research that focuses explicitly on Black men and health remains limited (Gilbert et al. 2016). This study draws on qualitative data in order to explore the health of Black men through the lens of food and eating as shaped by Black masculine identity. Adherence to traditional forms of masculinity, as mentioned previously, can have negative impact on men’s health [2,33]. However, research on faith has shown that it can shape health for the better. Faith is a part of Black communities in general [15,16] and Black masculine identity in particular [7] making faith a useful avenue for the study of Black men, health, and well-being. This paper focuses specifically on the experiences of Black men who identify as Christian or hold belief in a secular higher power. Interviews with men of different faith-based backgrounds would likely produce different findings and should be considered for future research.

## 2. Methods

### 2.1. Data Collection and Recruitment

The data for this study draws from a larger project, “Food, Social Relationships and Health Habits” study, which consisted of data gathered from both Black women and men. The “Food, Social Relationships and Health Habits” study was conducted as a follow-up study to a larger project as a means of generating additional interviews with Black participants. The larger study from which the data for this smaller study was drawn from sought to center the experiences of Black women in particular (n = 10). However, the narrative data generated from the Black male participants was too rich to ignore and thus lead to the development of this current study, which highlights the experiences of the five Black men interviewed for the larger project.

Data collection took place between September 2015 and September 2016. In-depth interviews were the primary data collection method for this study. In-depth interviews were selected as the primary data collection method insofar as they are useful for gaining insight into individual lived experiences [34,35]. Respondents were included in the study if they identified as Black, were between 30 and 60 years old, and had completed at least a bachelor’s degree at the time of the interview. Though other measures of social class such as income and wealth matter, education was the most accessible screening tool for recruitment for this particular study, which is in line with previous research [28]. Furthermore, as this particular study focused on the lived experiences of social relationships and food habits, education is a salient measure of social class insofar as education has the potential to “reflect tastes” [28] including what people consume.

Participants for this study were recruited through various means. Specifically, flyers, and other calls for participation were sent out to local community and professional organizations. Email listservs were utilized, in addition to active Black professional and community organizations on Facebook and LinkedIn. Remaining participants were reached using snowball sampling techniques. The “Food, Social Relationships, and Health Habits,” study generated 15 in-depth interviews with college educated Black women (n = 10), and men (n = 5). This particular study focuses on the narrative from the Black male respondents.

Each interview was conducted in a location decided on by the participant, including, cafes, restaurants, campus offices, and participants community club houses. On average, the interviews lasted 1.75 h and were digitally recorded. The interviews were later transcribed by the author. The author, a Black woman graduate student (at the time of data collection) conducted these interviews independently. Though the evidence on same race interviewing as a technique is conflicting and may not always be appropriate [36]; for the purpose of this study the author believes that it contributed to the narratives that the respondents shared. Simultaneously, the author recognizes that as a Black woman interviewing Black men, gender too may have played a role what was shared. Nevertheless, qualitative data offers insight into the “meanings, concepts, definitions, characteristics, metaphors, symbols, and descriptions” of people as well as things [37] (p. 3). The purpose of this particular study for instance, is to focus on the meanings and descriptions that Black men use when discussing their intentions to engage in health promoting behaviors, rather than quantify their overall experiences. That said, this study does not aim to be generalizable but rather to set a theoretical precedent for the examination of Black men’s experiences of food and eating as tied to their faith-based experiences. Beyond explicit denominational calls to engage in specific food behaviors during specific holiday’s, for instance, the participants in this study describe they ways in which their experiences of faith in general motivated their individual experiences of food and eating.

### 2.2. Sample

A total of 15 interviews were conducted between 2015 and 2016 with both Black women and men, 10 and 5, respectively, as a follow up study to a larger project that included data from Black women, Black men, as well as white women and white men. All of the participants in each study held at least a bachelor’s degree at the time of the interview. Education, holding at least a bachelor’s degree at the time of the interview, was used a marker of middle-class status for the participants within both sources of data collection—though the data for this particular project stem only from the smaller study conducted between 2015 and 2016. Measuring middle-class status among Black people in America varies widely, and is dependent on the mechanisms (e.g., income, occupation, education) used. Since Black women and men do not receive salaries or employment opportunities that necessarily match their educational background due to issues of racial discrimination [29,38] education on its own may be a better measure of middle-class status [29]. It is for reasons such as salary and occupational discrimination that this study drew on education as a marker of middle-class status for the participants. The data for this study, as mentioned previously, were collected as a follow up to a larger study that included interviews with Black women, Black men, and white women and white men; thus, educational attainment was the closest thing to an equalizer that could be used within the demographic parameters of the study in order to produce potentially similar access to food and social relationships (the focus of the larger study from which the data for this particular project stems).

Of the Black men in this study, two of the five respondents held graduate degrees, and the majority were single at the time of the interview (n = 3), with the remainder being divorced (n = 2). The median income of the Black men in this study was $50,000 for the four reporting Black male respondents. This is $9630 more than the median income of Black male full-time, year-round workers in the US [39]. The male respondents in this study were an average of 40 years old. Two of the respondents had children while the remaining three did not, and all but one of the male participants worked a full-time job.

### 2.3. Analysis

This study utilized the general theoretical assumptions of grounded theory as described by Charmaz (2014) as a means of moving beyond description to the development of theoretical insight derived from the respondents lived experiences. Each interview was read closely by the author multiple times in order to ensure familiarity with the data. The next step included a line-by-line coding approach as a means of identifying potential codes for in-depth analysis. Next, focused coding was used in order to assess the initial codes for analytic potential, while developing categories for later analysis. These categories were tied to religious/spiritual experiences and food (e.g., doctrine, fasting, etc.). Lastly, the author engaged in a process of theoretical coding in order to scrutinize how the analytic categories related to one another on a theoretical level, and how they were further connected to the literature on social relationships, faith, and the experiences of food and eating.

## 3. Findings

The narrative accounts detailed below are an attempt to offer qualitative insight into the ways in which Black men experience food and eating through the lens of faith as a dominant aspect of Black masculinity. Black masculinity has been discussed in a myriad of ways over time, from troublesome and deviant [23] to “inauthentic” and “performative” [7,19], to more complex discussions that incorporate several characteristics including faith driven, community focused, and survival-overcoming (resilience) [7]. The findings of this study suggest that Black middle-class men (as tied to having at least a bachelor’s degree) engaged in fasting as a means of promoting health from two faith-based spaces: (1) religious/spiritual as tied to God and (2) a belief in a higher power and community involvement. One of the unique aspects of these findings is that while the majority of men identify as Christian through the use of their reliance on Biblical texts, the food and eating behaviors that they engage in are not directly tied to denominational proscriptions but rather their own individual interpretations and desires to engage in health based behaviors.


*Fasting for God: “So, I turned to the Lord God and pleaded with him in prayer and petition, in fasting”*
*—Daniel 9:3*

The Daniel Fast, based on the biblical chapters of Daniel 1–10, is a fast that requires the elimination of commonly enjoyed foods for a period of twenty days as an act designed to bring the individual closer to God. Specifically, the faster is restricted from eating dairy, all meat, sugar (all forms of sweetener), yeast, refined and processed foods, deep fried food, and solid fats [40]. Thus, the faster is left with fruits, vegetables, whole grains, legumes, nuts, seeds, and oils exclusively for the twenty-day period. For Daniel, in the biblical sense, the fast was not about denying one’s self foods for foods sake but rather an opportunity to showcase one’s hunger for spiritual foods above the physical. The Daniel fast, named after the biblical text, then, is not a means to emulate the exact menu to which Daniel himself was limited but rather to imitate the spirit in which Daniel fasted [40]. Though the main purpose behind the fast was to move closer to God on a spiritual level, the food changes that are made shape health outcomes including weight loss and a higher intake of fruit and vegetables among participants. Research notes that many Americans do not meet the dietary guidelines for fruit and vegetable intake and that this is also tied to race and ethnicity [41]. Though we know the benefits of engaging in increased consumption of fruit and vegetables, not many people do so [41]. There are several factors that “contribute to this outcome and vary by social and situational family factors” [41], (p. 8).

One of the factors that contributes to such outcomes is race. Specifically, Black Americans tend to consume fewer fruits and vegetables per day [41]. Consumption of fruits and vegetables is tied as much to race as it is to access [41]; however, for Black Americans who have higher rates of mortality and suffer from other health disparities [8], engaging in activities that increase their fruit and vegetable intake would be beneficial to their overall health and well-being. For Jackson, a 34-year old Black man, the Daniel fast was a means for him to connect not only with his body but also with God. He stated,

“Yeah, Daniel from the Bible. You actually go 21 days only eating and drinking restricted things. You do it at the beginning of the year and that’s a sacrifice to God, showing Him that you’re willing to give these things up for Him. During the fast, you’re allowed to eat any fruits and vegetables, pretty much anything that grows from the ground. But no meats, no seafood, no seasonings other than salt and pepper. No dairy. You’re pretty much restricted for 21 days. I did it last year and it worked pretty well.”

For Jackson, his increase in fruits and vegetables and the elimination of fats and sugars resulted in weight loss, which he viewed as desirable. Jackson continued,

“I did it this year and I actually lost 18 pounds in 21 days. I can honestly say that I felt great during those days…my weight, I felt great. I even bought a smaller pair of jeans.”

Engaging in the practice of fasting was an opportunity to connect with his body, as well as to develop his connection to God. Jackson noted that other than the time he spent doing the Daniel’s fast, his eating habits were not necessarily the best. Specifically, Jackson stated,

“Yeah, I don’t eat healthy, but I make sure I work out, make sure I stay fit so that I don’t become obese. You know it reflects your overall health, and all aspects of your life. It really restricts you from doing things.”

Being obese, for Jackson was a health outcome that he tried to stave off with exercise and an increase in fruits and vegetables for at least one 21-day period throughout the year—the Daniel Fast. Diets that are high in fat but low in fiber are associated with numerous negative health outcomes ranging from certain cancers, to strokes, to heart disease [41] whereas high fruit and vegetable intake have been shown to act as a protective force against such outcomes [41]. By engaging in the *Daniel Fast* at least once a year, Jackson at least engaged in health promoting behaviors once a year. Faith as an aspect of Black masculinity in Jackson’s case produced at least one health promoting behavior—a yearly increase in fruit and vegetables.

Bryant, 41 and divorced, also noted that his faith played a role in shaping his food and heating habits throughout the year. Bryant noted that he was spiritually affected what and how he eats,

“…So, a lot of things that I’ve eaten, I have to stop. I’m trying to give up pork…someone messed around and showed me a picture of a baby lamb and that when I realized what it was…”

Bryant’s belief in God led him to make changes in his diet, and though he did not practice the *Daniel Fast* as Jackson did, he did find comfort in the practice of fasting. Bryant noted,

“Fasting, cleansing, spirituality…What I notice is that when I’m fasting, I don’t get headaches or hunger pain, which further empowers me and proves to be that there is evidence of all of that above. God sustains me when I’m hungry. It might be a placebo, but it’s God’s placebo. My spirituality has shown me that I have to stop ingesting things I know are bad for me.”

For Bryant this meant limiting the types of meats and other foods he put into his body as a means of preserving his health in order to adhere to his belief in God. For two of the Black men in my sample of 5, faith was based on religion/spirituality and impacted the ways that they engaged in the practice of fasting; faith as spiritual humanism was an intricate measure in Hunter and Davis’ (1992; 1994) conceptualization of Black masculinity. Nevertheless, participants engagement in fasting was complex, insofar as faith was multifaceted. To that end, faith for the participants in this study was not limited to the scope of religion/spirituality. For one participant, faith was community based and a means to bring health-based practices to their Black community—as a function of family within Hunter and Davis’ (1992; 1994) ideal of Black masculinity.


*The Black Health Challenge: “Try to do something good for your people—something difficult…if a man is poor, help him. Give him and his family food…”*
*—From a Winnebago lesson 1700s*

In addition to fasting as a practice of religious/spiritual engagement, Black men in this sample approached fasting as a mechanism of faith within the context of their community—specifically, the Black community of a local large southwestern town, as well as an extended online community. When it comes to conceptualizing Black masculine identities, one of the things that sets it apart from traditional notions of masculinity (e.g., success, aggression, the need to compete) is the culturally specific requirements like “cooperation, promotion of group and group survival” [25,26]. In that vein, The Black Health Challenge was taken up as a platform within the local large southwestern town mentioned above and is supported by an online community known as the Black Health support group—a private Facebook group only open to those participating in the challenge. The purpose of the group is to provide virtual, consistent support in the form of recipes, exercise exchanges, and motivational comments for members of this (Black) community in an effort to promote health and well-being. The Facebook support group is comprised of people from across the country according to Andre (33 years old) and offers a space for community connection as they work toward promoting the health of those who identify as Black/African American. Andre a graduate student said when we met in his office during his lunch break,

“…you see me eating peanuts. I’m actually in the middle of a fast, a fruit and vegetable fast. I started a fruit and vegetable fast for fourteen days, nothing but fruits and vegetables for fourteen days…we also began incorporating exercising, so this, I started it, but I got other Black people involved—it’s called the Black Health Challenge. It’s something I’ve done over the last six years, but I really just started getting other people involved in it the last two years or so...I do it every four months…Bringing others into it, I try to do it every two years. I have a friend, he moved back to Tampa, but me and him, we’re both farmers of The Black Health Challenge.”

Andre continued to discuss the challenge as a collective community of the individuals who identified as Black and/or African American. He also discussed his role in the group as being connected to farming and his love of gardening. Andre stated,

“I grow a lot of food. It’s one of my goals to be self-sufficient, anyway that I can, but particularly being able to get to the point where I do not have to go to the grocery store for very many things.”

Andre, though not necessarily invoking God as a higher power, did discuss faith as follows,

“I consider religion or even spirituality as your belief system, so the things you do in your life, the value system that you have…A big part of my value system, at least now, is healthy eating, wellness, and being holistic with your health…Um a big part of that is the community that you’re raised in, the family that you’re raised up in, and all these things that you’re a part of.”

For Andre, organized religion was connected to his past. In reflecting upon that past, he felt that organized religion was associated predominately with foods associated with “poor health.” He recalled,

“One thing I remember was the chicken that used to come about 3:30 pm, that’s when you knew [church] was about to be done, you’d smell the chicken. That fried chicken…you start to understand, really…I went home for my aunt’s funeral and, the same thing, they had chicken and what not, and it brought me back to my childhood, and I realized they had the same diets. Even though I had progressed and moved on with the consciousness, they hadn’t. So, who you are around, and those habits that you have really shape the things you are going to do later, unless something happens, something big happens to change that.”

For Andre, that growth came from the new relationships he was forming, romantic and otherwise, relationships that led him and others to the Black Health Challenge. Though chicken has long been a staple of the Black church community (Williams-Forson 2003), for Andre, it was a relic of the past that contributed to conditions such as diabetes and heart disease that ravaged Black communities. In Black church tradition, chicken or “the gospel bird” (Williams-Forson 2003) was and remains central to the religious experience. For instance, if the reverend were to come to a parishioner’s house for dinner, they would receive the “big piece of chicken” (Williams-Forson 2003). Usually fried but sometimes baked and served in other forms, chicken past and present is central to Black religious traditions (Williams-Forson 2003). For Andre, however, while it was “tradition[al]” and tied to “community” chicken in general, and fried chicken in particular, was responsible (at least in part) for the Black community’s poorer health outcomes.

Though not engaging in the practice of fasting, another of my male participants (n = 5), discussed religious spaces in particular as contributing to the downfall of Black health overall. Samson, a 60-year old man argued,

“I don’t have to go to Sunday dinner and I’m ok with that. I have gone to many church functions, I’ve gone to breakfasts at churches because I’m a health advocate, sometimes I eat the meals and sometimes I don’t. I’ve gotten very good at showing up, socializing, without having to eat a big pile of whatever it is they’re eating and its ok.”

For both Samson and Andre, church and religion were a source of negativity and poor health consciousness, while both Bryant and Jackson looked at organized religion in general, and connotations of Christianity in particular, as spaces in which they could better their health outcomes. Nevertheless, three of the four Black men who discussed fasting and faith (in all of its variations) viewed fasting as a form of health promotion.

## 4. Discussion

Faith as an aspect of Black masculinity plays a role in shaping the ways in which Black men engage with food and eating. Specifically, the Black men in my sample felt that their faith led them to fast as a means of health promotion, as well as connection to a power greater than themselves. As articulated in the study conducted by Hammond and Mattis (2005), faith was connected to God, a higher power, spirit, religious texts, or leaders; all the men in this study felt some form of faith. For Black men, faith then operates as a mechanism that facilitates positive health outcomes. Black men consistently have poorer health outcomes than both Black women and men of other racial/ethnic groups [8,9]. Specifically, in terms of life expectancy, Black men often have the highest rates of mortality due not only to health issues such as diabetes, prostate cancer, and hypertension [8,10] but also violence in the form of homicide (particularly among younger Black men) [8]. Many of these experiences are tied to Black men’s exposure to stressors from discriminatory and racist policies and practices within the US. Black masculinity as discussed by Hunter and Davis (1992, 1994) focuses on aspects such as pride, family, self-determinism, and humanism—all of which differ dramatically from traditional forms of masculinity in general and the negative representations of Black masculine identity in particular. As the literature notes, those who draw on alternative forms of masculinity may have lower health risk than the traditionally masculine counterparts. That said, it is important to note the external realities (e.g., lack, structural racism, lack of returns on education, etc.) that may trap Black men in negative masculine behaviors, which may lead to poorer health outcomes.

### Limitations

While there are several assets to this study, the limitations must be acknowledged. First, the data is limited to five participants, as the data was collected as a part of a follow up study to a larger examination of social relationships and health habits (e.g., diet, sleep, exercise). The follow-up study was not designed to focus on the experiences of Black men; however, the narratives generated from the study provide insight into a particular experience of faith and food among Black men. Second, this study drew upon educational attainment as a marker of middle-class status. Though the participants in the larger studies were highly educated (specifically the Black women), it may be argued that other markers of social class such as income, prestige, or occupation may be better measures of middle-class status and thus could have been considered within this study. A third limitation of this study is its emphasis on Black men who identify as Christian (though non-denominational) and/or spiritual without any defined religion. The experiences of Black men who participate in different faiths, Judaism, Islam, or more strict Christian denominations would likely have different experiences of food and eating related to their faith-based systems. Nevertheless, the goal of this paper was not to create generalizations to a broader population but rather to highlight the ways in which faith—in all of its complexities—impacted the experiences of specific Black men’s engagement with health-based behaviors. The men in this study provide a theoretical foundation for future study of Black men’s health through the lens of food and eating as tied to faith.

## 5. Conclusions

Research on Black men’s health is growing, yet it still requires further examination. One of the ways to examine the health of Black men in particular is through the lens of faith as well as conceptualizations of Black masculinity. Black masculinity is complex and multifaceted, yet as alternative to traditional forms of masculinity, aspects of Black masculinity lend themselves to the pursuit of health promoting attitudes and behaviors. This study has provided narrative accounts of fasting via the lens of faith as an aspect of Black masculine identity. Further qualitative studies could draw on larger samples of Black men and consider the role of faith in multiple ways as means of digging deeper into the lived experiences of Black men with respect to health and masculinity.

This paper contributes to the literature on Black masculine identity as well as Black men’s potential health and well-being in the context of faith. Furthermore, this study pulls together threads that connect the experience of Black masculinity and faith as mechanisms of health promotion among a group that has the poorest health outcomes and yet receives the least attention—Black men [8]. Faith, in all of its forms was the driving force behind these men’s desires to pursue health promotion. Qualitative studies are needed, not only to document the phenomena of Black men’s experiences of health but also to highlight potential pathways for intervention in health focused programs geared towards Black men in particular. Qualitative data has the strength of digging deeply into the lived experiences of respondents and providing narrative accounts of their thoughts and feelings towards those experiences [34,35]. Future research would do well to examine the relationship between food and faith with respect to differing denominations within Christianity, as well as different faiths such as Islam and Judaism. Various faith backgrounds have different relationships to food and eating as tied to the specific guidelines of religious texts, and thus, their experiences would likely be tied to such mandates. The Black men in this study, while drawing on Biblical texts, did not engage in patterns of food and eating based on denominational backgrounds but rather through their own interpretations and desires to engage in health-based behaviors.

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
