# Peer review of "Eat to Live, Don’t Live to Eat: Black Men, Masculinity, Faith and Food"

_ijerph, 2020, doi:10.3390/ijerph17124264_

Round 1
Reviewer 1 Report
This is a timely paper given that we are in a pandemic, which is showing that there are marked differences in the way COVID-19 affects different communities, with Blacks being the worst off because of underlying conditions such as diabetes and obesity. Muh of this is due to eating habits. A study focusing on how Black men eat healthily is timely.
Having said this, eventhough the paper is well-written, I am quite disappointed at the very small number of participants...only 5 men! No generalizations can come from this.
I am also a bit worried about the implications of the study: can we now deduce that the more faith-oriented a Black man is the more likely he would eat healthily? Again, this number of 5 is too small to make any generalization.
A contrary hypothesis could be that actually sports is more of determining factor than faith/spirituality in determining whether a Blackman eats healthily or not. I think there are more healthy Black sportsmen that healthy religious Black men.
While the paper is also well-written, there is at least one typo in the concluding section of the paper.
On the whole, I am not very convinced about the link between faith and healthy eating...and would like that the author convinces me more.
Reviewer 2 Report
Your research is interesting and well supported by evidence. You have meticulously followed academic methodological research, although in doing that you were caught in the the trap of making repetitive statements and sentences. There are two main problems that you can readily address. The first is that your definition of "Black men" should be in the abstract as well as the place where you first use the term or phrase in the "Introduction" of the paper proper (page 2, line 44). You later gave the definition of term in lines 70-71, but it should come earlier. In the United States, there are Black men from Latin America, from the Caribbean, from Africa, and those who are native-born (not foreign-born, naturalized) Black citizens of the United States. You want to avoid that confusion in the inquisitive mind of an informed, critical reader by explaining the limitation placed on that term as an operative phrase in the paper. One also wonders why you did not use the current, more limiting term, African American, which you use on page 3, line 112. If the reason for not using the term, "African American," is that you want to make stark racial comparison of White and Black, then state it so when you first use the term.
The second problem to address is the presumption/assumption that every Black man or African American male is a Christian or of the Christian faith. There are Black men who are of the Islamic faith or of the Nation of Islam, and those who practise Santeria that is derivative of African traditional religion. There are also Black Jews. Even Christianity, as a religion, has many denominations. You should break it down further to show that your research is focused on the Black Church; that would place the "Daniel Fast" and the "gospel bird" (chicken) in the right context of faith and health.
You should proofread your paper carefully to correct the following:
page 1, line 40: delete "of" and replace with "that" to read "...the role that social..."
line 43: change "construction" to construct.
page 2, line 66: change the plural verb "shape" to singular, shapes.
line 74: no need to repeat "this paper;" replace the phrase with the pronoun, it. Thus, the sentence will begin with "Next it examines..."
line 94: change "push" to pushes since the verb refers to "Research."
page 3, lines 95-96: place a comma after "[22]," and place a period after "[7]."
line 97: change "paint" to paints since the verb refers to "Research."
line 103: place a comma after "study" and after "too.
line 105: change "with" to within.
line 106: change "doesn't" to does not to make it formal.
line 114: place a comma after society.
line 119: place a comma after "men" and after "general" to read "Men, in general,..."
line 124: remove the brackets after "counterparts []" since nothing is placed within them.
line 124: delete "However" and begin the sentence with Though.
line 125:insert "in" to read "participate in negative..."
line 126: delete "however."
line 129: insert "as" for the phrase to read "(as well as Black women)"
line 130: change "negative" to negatively.
line 134: change "that as a result" to "that are a result..."
line 139: change "continue" to continues since it is linked to study.
page 4, line 145: place a comma after "decades,"
line 148: insert "as" after "such" to read "such as a lowered..."
line 154: change "different than..." to read "different from..."
line 161: place a comma before and after "too" to read "food behaviors, too, matter..."
line 165: change "attitudes and health-based" to "attitudes toward health-based..."
line 167: change "faith relationship with faith and food" to "faith relationship with food"
line 174: place a comma to read "forbidden, how..."
line 176: change "becomes in more..." to "becomes more..."
line 177: change "many was..." to "many ways..."
line 181: delete the comma after "in nature"
line 185: delete the comma after "worship"
line 185: delete the apostrophe in "spaces' faith..." and replace with a comma to read "spaces, faith..."
line 192: change semicolon to comma.
line 194: change "Back masculine" to "Black masculine."
page 5, line 195: "can have a negative..."
line 197: insert "consciousness" to read "Black communities' consciousness in general..."
line 198: insert a comma after "particular [7], making..." and use a pronoun "it" to replace "faith" to read "making it a useful..."
line 205: delete "In-depth interviews were selected as the primary data collection method because they..." and replace with "This is because in-depth interviews are useful for..."
line 207: insert a comma after "old,"
line 215: insert a comma after "participation,..."
line 218: insert a hyphen to read "college-educated..."
line 225: change semi-colon after "[36];" to a comma.
line 227: place comma to read "gender, too, may..."
line 227: insert "in" after "role" to read "role in what..."
line 231: insert hyphen in "health-promoting behaviors..."
line 234: delete "of" from "2016 of with..."
line 238: insert "as" to read "used as a marker..."
line 239: change "stem" to "stems." It is an acceptable common practice to use a singular verb for data.
line 243: insert a comma after "[29, 38],"
page 6, line 246: change "were" to "was"; place a hyphen in "follow-up"
line 247: change comma after "white men," to a semi-colon to read "white men;"
line 269: Incomplete sentence. Complete it or delete it.
line 274: replace semi-colon after "[7,19];" with a comma. After the comma, insert "and" before "to more..."
line 276: place a period after "[7]."
line 279: delete one of the two periods after "involvement.."
line 281: delete the dash "-" at the end of the quote; or move it to line 282 and place it in front and next to "- Daniel 9:3"
line 285: delete the colloquial word, "faster,'" and replace it with "person fasting."
line 287: change "the faster" to "the person fasting" or "the one fasting."
line 288: change "foods" to "food" as the more formal plural of food.
line 289: change "foods sake" to "food's sake" and change "spiritual foods" to "spiritual food."
page 7, line 292: insert "is" after "fast."
line 293: change "fruit" to "fruits"; do the same on line 296.
line 298: place a period after "p.8]."
line 300: place a period after "[41]."
line 301: delete semi-colon after "[41];" and replace it with with a period; capitalize "H' in "however" to begin the sentence.
line 302: move the comma after "[8] ," closer to the bracket.
lines 306-311: Indent this long quote instead of using quotation marks.
lines 314-315: This is a short quote. You don't need a paragraph for it. Let the quote begin after "Jackson continued," on line 313.
line 318: place a comma after "fast" and delete "that"; change "weren't" to "were not."
lines 319-321: This is a short quote that doesn't need a new paragraph. Begin the quote from line 318.
line 322: place a comma after "Jackson,"
line 323: make "in fruits" corrected to "in eating fruits" and do not italicize "the Daniel Fast."
line 325: insert "and diabetes" after "[41]"
line 326: change "have" to "has" since the verb is linked to "intake."
lines 326-327: don't italicize "Daniel Fast."
line 327: place a hyphen in "health-promoting behaviors..."
line 328: make it "Jackson's case"; "health-promoting behavior"
line 330: begin the sentence with "Another participant, Bryant..."; change "heating" to "eating"
line 331: "spiritually affected by what and how..."
lines 330-335: make it one paragraph.
line 335: : don't italicize "Daniel Fast"
lines 336-339: indent the four lines instead of using quotation marks
line 340: "For Bryant, this meant..."; change "foods" to "food"
line 342: replace comma after "fasting," with a semi-colon;
page 8, line 344: "participants' engagement"
line 358: delete the comma after "group survival" [25, 26].
line 359: delete "which"
line 360: "mentioned"
line 365: place a comma after "country," and another after "...old),"
line 366: place a comma after "Andre,..."
line 367: place a comma after "student," and replace the comma after "break" with colon:
lines 368-374: indent this long quote and remove quotation marks
lines 375-379: make these two short paragraphs one; use closing quotation mark at the end of line 379
lines 388-394 (on page 8): indent this long quote and remove quotation marks
line 413: delete comma after "church," and change "was" to "were."
line 424: change "facilitate" to "facilitates"
line 426: place a period after "[8,9]."
line 427: insert "and" after "cancer,and hypertension..."
line 435: delete "lack," in "(e.g.lack,..."
line 439: insert a hyphen in "follow-up..."
line 443: delete "more"
line 444: replace "should've" with "should have"
line 449: Replace the first sentence with this: "Since research on Black men's health is still growing, the topic requires further examination."
line 453: insert a hyphen in "health-promoting attitudes..."
line 461: delete "is" in "is has..."
line 462: place a comma after "...forms,"
line 464: place a hyphen in "health-focused..."
line 468: place a period at the end of the sentence.
line 469: remove "s" from "authors" (author) and add "s" to "declare" (declares)
For References:lines 471 and 472; 494 and 495; 509 and 510; 515 and 516; 517 and 518; 525 and 526; 538 and 539; 540 and 541; 545; 547 and 548; 553: These are book references. Check them for accuracy, especially the order of placement of city/place of publication, and the name of publisher. Here is an example (lines 471 and 472, the first on your list. You numbered it 1. 1. Delete one of the two "1." numbers.): Connell, R. Gender and Power: Society, the Person and Sexual Politics. Palo Alto, CA: Stanford University Press, 1987.
Round 2
Reviewer 1 Report
I believe the paper should now be published, though I still have reservations about the small number of participant: n = 5. This is not good for generalizations at all, and the author must make this clear in the limitations section!